# Long Non-Coding RNA LOC401312 Induces Radiosensitivity Through Upregulation of CPS1 in Non-Small Cell Lung Cancer

**DOI:** 10.3390/ijms26125865

**Published:** 2025-06-19

**Authors:** Zhengyue Cao, Tiantian Wang, Fumin Tai, Rui Zhai, Hujie Li, Jingjing Li, Shensi Xiang, Huiying Gao, Xiaofei Zheng, Changyan Li

**Affiliations:** State Key Laboratory of Proteomics, National Center for Protein Sciences (Beijing), Beijing Institute of Radiation Medicine, Beijing 100850, China; caozhengyue29@gmail.com (Z.C.); amtianwtt@163.com (T.W.); taifm810@163.com (F.T.); lihujie0309@163.com (H.L.); xfzheng100@126.com (X.Z.)

**Keywords:** CRISPRa screening, long noncoding RNA, LOC401312, CPS1, ionizing radiation sensitivity regulation

## Abstract

Long noncoding RNAs (lncRNAs), non-protein-coding transcripts exceeding 200 nucleotides, are critical regulators of gene expression through chromatin remodeling, transcriptional modulation, and post-transcriptional modifications. While ionizing radiation (IR) induces cellular damage through direct DNA breaks, reactive oxygen species (ROS)-mediated oxidative stress, and bystander effects, the functional involvement of lncRNAs in the radiation response remains incompletely characterized. Here, through genome-wide CRISPR activation (CRISPRa) screening in non-small cell lung cancer (NSCLC) cells, we identified LOC401312 as a novel radiosensitizing lncRNA, the stable overexpression of which significantly enhanced IR sensitivity. Transcriptomic profiling revealed that LOC401312 transcriptionally upregulates carbamoyl-phosphate synthase 1 (CPS1), a mitochondrial enzyme involved in pyrimidine biosynthesis. Notably, CPS1 overexpression recapitulated the radiosensitization phenotype observed with LOC401312 activation. Mechanistic investigations revealed that CPS1 suppresses the phosphorylation of ATM kinase (Ser1981) protein, which is a key mediator of DNA damage checkpoint activation. This study established the LOC401312–CPS1–ATM axis as a previously unrecognized regulatory network governing radiation sensitivity, highlighting the potential of lncRNA-directed metabolic rewiring to impair DNA repair fidelity. Our findings not only expand the functional landscape of lncRNAs in DNA damage response but also provide a therapeutic rationale for targeting the LOC401312–CPS1 axis to improve radiotherapy efficacy in NSCLC.

## 1. Introduction

The Human Genome Project (HGP) and ENCODE revealed ~80% of the genome is transcriptionally active, yet only 2% of it encodes proteins, with the majority comprising non-coding RNAs (ncRNAs) [1,2,3]. lncRNAs, once deemed “junk RNA”, are now recognized as multifunctional regulators of cellular differentiation, development, and stress responses [2,4]. They modulate the three-dimensional chromatin architecture [5], stabilize telomeres [6,7], remodel chromatin via histone modifications [8], regulate imprinting genes [9], and control transcription through promoter-associated (paRNAs) [10] or enhancer RNAs (eRNAs) [11]. Despite this research progress, >70% of lncRNAs lack functional annotation, with many uncharacterized due to tissue-specific expression. Critical gaps persist in defining their structural motifs, dynamic interactions, and physiological roles beyond model systems. Their functional redundancy, context-dependent activity, and spatiotemporal regulatory hierarchies remain poorly resolved. Deciphering lncRNA biology requires integrating advanced tools to map their mechanistic diversity and organismal impact. lncRNAs, recognized as critical regulatory molecules in tumor biology, have emerged as pivotal modulators of cancer therapeutics [12]; however, their functional roles and mechanistic contributions to radiotherapy responses remain poorly characterized, necessitating systematic investigations to delineate their therapeutic potential in radiation oncology and elucidate their regulatory dynamics in modulating tumor radiosensitivity.

CRISPR-based genome engineering has emerged as a cornerstone technology in functional genomics research due to its high-precision genome editing capabilities, enabling large-scale functional screens through molecular engineering innovations. The catalytically deactivated dCas9 variant has been incorporated into modular platforms for locus-specific epigenetic reprogramming, retaining sequence-specific DNA anchoring while acquiring transcriptional control through effector domain fusions (e.g., VP64 transactivation complexes) [13]. CRISPRa systems epitomize this technological progression, deploying dCas9-transactivator chimeras guided by sgRNA constellations to achieve transcriptional amplification at promoter-proximal genomic coordinates [14,15]. Parallel advancements in CRISPR screening methodologies employ multiplexed sgRNA libraries for systematic interrogation of gene regulatory networks, where genome-scale or pathway-focused guide RNA collections enable combinatorial genome perturbations coupled with deep phenotyping and next-generation sequencing (NGS)-based deconvolution to resolve genotype–phenotype correlations. Continuous methodological refinements have spawned applications ranging from genome-wide transcriptional activation mapping—revealing histone acetyltransferase (HAT) inhibitor complexes as HIV-1 proviral integration modulators [16]—to precision oncology investigations where CRISPRa screens delineated metabolic adaptation networks underlying metformin resistance in prostate cancer [17]. This proven capacity for functional lncRNA annotation establishes CRISPRa as a transformative framework for decrypting radiation-inducible non-coding RNA circuits, particularly in mapping therapeutically actionable lncRNA nodes within ionizing radiation response cascades.

Ionizing radiation exerts its biological impact through two primary mechanisms: direct ionization of molecular targets such as DNA and proteins and indirect effects mediated by reactive oxygen species (ROS) generation, thereby inducing structural alterations and functional dysregulation in these critical biomolecules [18,19,20]. Emerging evidence elucidates an expanded regulatory architecture of ionizing radiation responses, wherein lncRNAs serve as bifunctional mediators, concurrently operating as molecular rheostats modulating radiosensitivity through epigenetic coordination of DNA repair machinery while exhibiting biomarker potential via radiation dose–responsive transcriptional dynamics [21,22,23,24]. Mechanistic investigations in oncological contexts reveal lncRNAs orchestrate cellular radiation responses through metabolic network crosstalk, with urothelial cancer associated 1 (UCA1) constituting a prototypical regulator that augments radiosensitivity in cervical carcinoma models through targeted suppression of hexokinase 2 (HK2), the pivotal glycolytic gatekeeper enzyme [25]. The growth arrest-specific 5 (GAS5) lncRNA augments radiation responsiveness through inhibition of miR-21 and activation of the PI3K/Akt signaling cascade [26]. Conversely, LINC01134 establishes radiation defense systems in hepatocellular carcinoma via kinase rewiring of MAPK-mediated cell cycle checkpoints, clinically correlating with acquired radioresistance phenotypes [27]. This regulatory cartography establishes lncRNAs as nodal regulators operating across the radiation biology continuum, from real-time dosimetric biomarkers to druggable targets within therapeutic resistance networks.

Despite growing recognition of non-coding RNAs as master regulators of radiation response pathways, the systematic identification and mechanistic characterization of functionally significant lncRNAs in radiation biology remains an unresolved challenge. In this study, we utilized CRISPRa library-based high-throughput screening and identified LOC401312 as a central regulator of radiosensitivity in NSCLC. Mechanistically, we demonstrated that LOC401312 functions as a transcriptional enhancer of CPS1, thereby revealing an unexpected connection between urea cycle metabolism and radiation response in NSCLC pathogenesis. Our study identified CPS1 as a critical radiosensitizing factor in lung cancer. It physiologically functions as the rate-limiting enzyme in the urea cycle’s initial reaction mediating detoxification of the genotoxic metabolite ammonia, with established pathophysiological relevance to diverse human diseases including cardiovascular disorders and malignancies [28].

## 2. Results

### 2.1. Systematic Identification of Radiation-Responsive lncRNAs via CRISPRa Screening in NSCLC

To investigate lncRNAs with critical roles in the ionizing radiation response, we performed functional screening of 4766 lncRNAs expressed in non-small cell lung cancer using a CRISPRa non-coding RNA library containing 6 sgRNAs per lncRNA and 1884 control sgRNAs packaged into puromycin-resistant lentiviral vectors. Following lentiviral transduction of A549-dCas9-VPR cells at a multiplicity of infection (MOI) of 0.1, we performed 4-day puromycin selection to establish polyclonal populations with single sgRNA integrations. Cells were then subjected to γ-irradiation (8Gy, ^60^Co source), with genomic DNA extracted from irradiated and non-irradiated control cells at day 14 post-treatment for sgRNA abundance quantification through next-generation sequencing (Figure 1A). Two independent biological replicates of the CRISPRa screening experiments were performed. Bioinformatic interrogation of sgRNA distribution patterns revealed distinct radiation-responsive candidates: enriched sgRNAs corresponded to radioresistance-associated lncRNAs, while depleted sgRNAs indicated radiosensitizing targets, collectively identifying 1329 radioresistant and 177 radiosensitive lncRNAs (Figure 1B). Protein interaction networks of these lncRNAs analyzed via LncSEA (http://bio.liclab.net/LncSEA, accessed on 1 November 2021) demonstrated significant enrichment in PI3K–Akt and MAPK signaling pathways (Figure 1C), consistent with established lncRNA-mediated radiation response mechanisms. The beta score ranking and differential gene visualization revealed radiosensitivity-associated lncRNAs identified through our screening, with LOC401312 emerging as a prioritized candidate (Figure 1D). Multi-sgRNA correlation validation across prioritized lncRNAs demonstrated robust consistency (Figure 1E), substantiating the reliability of our screening results. Further clinical correlation analysis using GEPIA2 (http://gepia2.cancer-pku.cn, accessed on 10 June 2025) revealed upregulation of LOC401312 in lung adenocarcinoma (LUAD) and squamous cell carcinoma (LUSC) tissues compared to adjacent normal controls (Figure 1F), identifying this significantly enriched radiosensitizing lncRNA with established disease relevance as a foremost candidate for mechanistic dissection.

### 2.2. LOC401312 Validated as a Radiosensitizing lncRNA in Lung Cancer Cells

LOC401312, a five-exon lncRNA transcribed from the STEAP1B locus on chromosome 7, was functionally characterized through overexpression in A549 and H460 lung cancer cells, with qRT-PCR validation confirming robust transcriptional activation efficiency (Figure 2A). Radiation sensitivity analyses of cells were performed via CCK-8 proliferation assays and clonogenic survival tests following 8 Gy irradiation. Overexpression of LOC401312 in A549 cells significantly reduced cellular proliferation by 21.7% (*p* < 0.0001) at 72 h post 8 Gy ionizing radiation compared to controls (Figure 2B), while colony formation assays demonstrated a marked decline in the colony-forming capacity of LOC401312-overexpressing cells (Figure 2C). At 8 Gy irradiation, which exceeds the threshold of cellular repair, A549 cells exhibited complete loss of viability regardless of experimental grouping (vector control vs. LOC401312-overexpression cohorts), and no significant difference was observed between LOC401312-overexpressing and control groups (Figure 2D). Cross-validation in the H460 lung cancer cell line confirmed that LOC401312 overexpression similarly enhanced cellular sensitivity to ionizing radiation. To evaluate DNA damage repair modulation, phosphorylation of histone H2AX at serine 139 (γH2AX), an established molecular marker for double-strand break (DSB) formation and repair progression, was analyzed at 2 and 8 h post-2 Gy γ-irradiation. Cells were fixed, stained with γH2AX-specific antibodies, counterstained with DAPI, and subjected to foci quantification (>20 foci/nucleus defined as positive) (Figure 2E). In control cells, γH2AX foci formation was sharply increased at 2 h post-irradiation, declined by 8 h, and nearly resolved at 24 h, indicating efficient DNA damage repair. In contrast, LOC401312-overexpressing cells exhibited significantly elevated γH2AX foci counts compared to controls, with sustained high levels persisting at 8 h and residual foci detectable at 24 h post-exposure, demonstrating that LOC401312 overexpression amplifies radiation-induced DNA damage and partially impairs repair (Figure 2F). These findings collectively demonstrate that LOC401312 overexpression amplifies radiation-induced DNA damage, suggesting its potential role in radiation sensitization through modulation of DNA damage response pathways in A549 cells.

### 2.3. Transcriptomic Profiling of LOC401312-Overexpressing A549 Cells Identifies CPS1 as a Functional Mediator of Radiation Sensitivity

Transcriptomic sequencing with three independent biological replicates comparing LOC401312-overexpressing versus control cells identified 205 upregulated and 110 downregulated genes (|log2(fold change)| > 1, adjusted *p*-value < 0.05) (Figure 3A). GO and KEGG enrichment analyses of significantly differentially expressed genes from transcriptomic sequencing revealed a predominant enrichment in amino acid metabolism pathways, providing critical clues for subsequent investigations. Convergent evidence from these analyses, combined with volcano plot characterization, collectively implicated CPS1 as a prioritized candidate (Figure 3B,C). Cross-validation in multiple lung carcinoma cell lines confirmed CPS1 as the most robustly induced target, where LOC401312 overexpression in A549 cells elevated CPS1 mRNA levels by 4.1-fold compared to empty vector controls (Figure 3D), with Western blot analysis further demonstrating concomitant upregulation of CPS1 protein expression (Figure 3E) and parallel results observed in the H460 lung cancer cell line, collectively highlighting consistent transcriptional activation and protein-level induction across models. Pharmacological inhibition of CPS1 enzymatic activity using the specific small-molecule inhibitor H3B-120 in LOC401312-overexpressing A549 cells significantly reversed the radiation sensitization phenotype (Figure 3F,G), demonstrating the functional dependency of LOC401312-mediated radiosensitization on CPS1 catalysis. These findings indicate that the radiation-sensitizing effects triggered by LOC401312 overexpression are mediated via CPS1 enzymatic activity.

### 2.4. CPS1 Overexpression Enhances Radiosensitivity in Lung Carcinoma Cells

To evaluate the role of CPS1 in LOC401312-induced radiation sensitization, stable CPS1-overexpressing cell lines were generated in A549 and H460 models via jetPRIME-mediated transfection with human CPS1 cDNA-containing plasmids. RT-qPCR analysis revealed that overexpression of CPS1 in A549 cells elevated its mRNA levels by 37.25-fold compared to mock controls (Figure 4A), with Western blot analysis confirming a marked increase in CPS1 protein expression (Figure 4B), while parallel experiments in H460 cells demonstrated a 27.88-fold upregulation of CPS1 mRNA and a corresponding significant protein-level induction. The proliferation assay demonstrated significantly impaired cellular proliferation in CPS1-overexpressing lung cancer cells compared to vector controls following 8 Gy ionizing radiation (Figure 4C). Clonogenic survival tests revealed markedly enhanced radiosensitivity in CPS1-overexpressing lines post-8 Gy irradiation, exhibiting 72.2% (*p* = 0.0002) and 52.7% (*p* = 0.0029) reductions in colony formation capacity for A549 and H460, respectively, compared to controls (Figure 4D,E). These findings indicate that CPS1 overexpression significantly enhances ionizing radiation sensitivity in both A549 and H460 cell lines. To directly validate CPS1’s role in NSCLC, we attempted to generate stable CPS1-knockout (KO) cell lines in the A549 and H460 models. Consistent with prior reports that CPS1 depletion suppresses nucleotide biosynthesis and cell growth [29], baseline proliferation was significantly inhibited in both lines. Consequently, establishing stable CPS1-KO pools proved infeasible under standard culture conditions. CPS1 protein abundance remained unaffected by ionizing radiation, showing no significant alterations in expression levels under basal conditions or at 8 or 24 h post-irradiation timepoints (Appendix A).

### 2.5. Co-Expression Network Analysis of CPS1 in Lung Cancer Identifies DNA Damage Repair Modulation as a Mechanism Governing Ionizing Radiation Sensitivity

Integrated analysis of CPS1 co-expression networks in TCGA lung cancer cohorts via cBioPortal identified 173 significantly correlated genes (Appendix A). Volcano plot analysis of co-expression profiles showed marked downregulation of radiation-associated genes ATM (Figure 5A), with ATM showing a log ratio of −0.23 (*p* = 0.308). Protein–protein interaction (PPI) network mapping revealed core hub genes enriched in DNA damage response (DDR) regulators including AKT1, ATM, and RAD50 (Figure 5B). GO analysis demonstrated significant enrichment in oxidative stress response, gamma radiation response, and ionizing radiation response pathways (Figure 5C). Western blot analysis revealed that CPS1-overexpressing A549 cells exhibited significant downregulation of ATM protein levels and reduced phosphorylation of ATM at Ser1981 (p-ATM) compared to vector controls at 2 h post-8 Gy irradiation, accompanied by elevated γH2AX accumulation (Figure 5D). By 24 h post-irradiation, sustained ATM/p-ATM suppression persisted in CPS1-overexpressing cells despite γH2AX levels declining to near-baseline thresholds (Figure 5E). In H460 cells, pronounced γH2AX elevation with minimal ATM reduction was observed alongside attenuated p-ATM post-irradiation (Appendix A). Notably, pharmacological inhibition of CPS1 enzymatic activity using H3B-120 prior to irradiation restored both ATM and p-ATM expression to baseline levels, confirming the functional dependency of this regulatory axis on CPS1 catalysis. Immunofluorescence analyses conducted at 2, 8, and 24 h post-ionizing radiation further corroborated the exacerbated DNA damage in CPS1-overexpressing A549 cells (Figure 5F,G). Comet assay analysis further revealed significantly higher tail moments in CPS1-overexpressing cells compared to vector-control counterparts following ionizing radiation exposure (Figure 5H), indicating elevated DNA strand break levels relative to control cells (Figure 5I). Additional investigations in CPS1-overexpressing A549 cells aiming to determine whether CPS1 regulates ATM levels via ROS-mediated mechanisms revealed no significant alterations in intracellular ROS levels upon CPS1 overexpression (Appendix A). Given the metabolic significance of CPS1 and its enzymatic product carbamoyl phosphate, we systematically investigated whether carbamoyl phosphate regulates ATM levels in A549 cells. Western blot analysis confirmed that treatment with varying concentrations of carbamoyl phosphate did not change the abundance of ATM protein (Appendix A). These findings delineate a novel urea cycle-dependent mechanism through which CPS1 metabolically regulates ATM kinase activity to influence DNA repair and radiation sensitivity in lung carcinoma cells.

## 3. Discussion

This study demonstrated that LOC401312, a radiosensitizing lncRNA identified through CRISPRa screening, enhances ionizing radiation sensitivity in lung cancer cells via the LOC401312–CPS1 transcriptional regulatory axis. Mechanistic investigations further revealed CPS1 as a putative modulator of the DDR pathway in lung carcinoma, orchestrating radiation-induced genomic instability through coordinated regulation of DDR effectors.

lncRNAs, representing the largest class of non-coding RNAs with over 90,000 annotated genes [30], play pivotal roles in transcriptional and post-transcriptional regulation through interactions with DNA, RNA, or proteins [31]. Emerging evidence implicates specific lncRNAs as promising radiation response modulators and biomarkers [32]. The radiosensitizing lncRNA LOC401312 identified in this study represents an uncharacterized lncRNA, with no prior reports linking it to ionizing radiation sensitivity regulation. Overexpression of LOC401312 enhanced radiation sensitivity in A549 and H460 cells, exacerbating early DNA damage severity, while the CPS1-specific inhibitor H3B-120 reversed this radiosensitizing effect. Preliminary mechanistic exploration revealed a partial overlap between microRNAs targeting CPS1 and those computationally predicted to interact with LOC401312, suggesting potential regulatory cross-talk that may guide future investigations into the LOC401312–CPS1 axis.

CPS1, the rate-limiting enzyme of the urea cycle, catalyzes the ATP-dependent condensation of bicarbonate, ammonia, and ATP into carbamoyl phosphate within hepatic mitochondrial matrices under physiological conditions [33]. As a central regulator of hepatic metabolic homeostasis, CPS1 not only mediates ammonia detoxification but also exhibits aberrant expression patterns correlating with chronic hepatitis C virus (HCV) progression and liver fibrosis staging [34]. Intriguingly, CPS1 displays tissue-specific oncogenicity: its overexpression in colorectal tumors of patients with a poor neoadjuvant chemoradiotherapy response is associated with reduced pathological complete remission rates and shortened progression-free survival [35], while hepatocellular carcinoma studies demonstrate that CPS1 deficiency enhances radiation resistance through c-Myc regulation [36]. Notably, our findings align with CPS1’s radiation-sensitizing role in lung cancer, where co-expression network analysis and post-irradiation Western blot assays revealed enzyme activity-dependent modulation of DNA damage repair and oxidative stress pathways. In our models, CPS1 overexpression induced significant downregulation of the DNA damage response mediator ATM, impairing radiation-induced phosphorylation of ATM at Ser1981 as demonstrated by Western blot analyses. ATM, a master regulator of DNA double-strand break repair, orchestrates the activation of cell cycle checkpoints, DNA repair cascades, and apoptotic pathways through phosphorylation of downstream targets including Chk2 and p53 [37,38,39]. The observed suppression of ATM expression in CPS1-overexpressing lung carcinoma cells creates a compounded DNA repair deficiency phenotype, mechanistically explaining the enhanced radiation sensitivity. The physical interaction between CPS1 and H2AX documented in the NCBI database implies that γH2AX upregulation induced by CPS1 overexpression may operate through regulatory pathways distinct from ATM downregulation, warranting further investigation into the mechanistic implications of this protein–protein interaction.

This study demonstrated that the urea cycle enzyme CPS1 functions as a DDR pathway modulator in lung cancer radiobiology. The identified LOC401312–CPS1 axis presents a therapeutic strategy to enhance radiosensitivity via targeted axis activation. CPS1 emerges as a dual-functional candidate in lung cancer patients, serving both as a potential diagnostic and prognostic biomarker [29,40,41,42] and a radiosensitizing therapeutic target to optimize radiotherapy efficacy through its novel role in metabolic regulation of DNA damage response pathways. Preliminary analyses integrating TCGA data tentatively propose that CPS1 in lung cancer might not only represent a candidate biomarker, as suggested by prior investigations, but could potentially correlate with radiotherapy efficacy and treatment-related prognostic outcomes, whereas the observed differential expression patterns of LOC401312 between histologically normal adjacent tissues and neoplastic lesions may hypothetically inform biomarker-driven approaches for refining therapeutic radiation protocols in clinical management. Our bioinformatic interrogation (TargetScan v8.0 and miRCODE) reveals putative miRNA regulators of CPS1 overlapping with the LOC401312-predicted targetome, including miR-101, miR-141, and miR-144; which exhibit computationally predicted binding affinities for CPS1 mRNA. These findings prompt the hypothesis that LOC401312 may regulate CPS1 expression through miRNA intermediaries, although functional validation would be warranted to mechanistically resolve this apparent regulatory triad—a critical avenue for subsequent investigation. While the current findings provide mechanistic insights into radiation response modulation, this study is inherently limited by the absence of in vivo validation through preclinical models or patient-derived xenograft systems; further investigations should delineate the molecular mechanisms underlying LOC401312-mediated CPS1 transcriptional regulation and explore urea cycle metabolite dynamics in clinically relevant radiation response models.

## 4. Materials and Methods

### 4.1. Cell Culture and Transfection

The A549 and H460 cell lines, obtained from the American Type Culture Collection (ATCC, Manassas, VA, USA), were cultured in their respective growth media: A549 cells were cultured in Dulbecco’s Modified Eagle Medium (DMEM; Gibco, Pleasanton, CA, USA) and H460 cells were cultured in Roswell Park Memorial Institute 1640 medium (RPMI 1640; Gibco, Pleasanton, CA, USA), both supplemented with 10% fetal bovine serum (FBS; Pan-Biotech, Adenbach, Germany) and 1% penicillin-streptomycin. Cell transfections were performed using the jetPRIME transfection reagent (Polyplus, Illkirch, France) according to the manufacturer’s protocol.

### 4.2. Infection of lncRNA Activation Libraries and Screening

Lentiviral CRISPR library transduction commenced at 80% confluence following ice-thawed viral preparation, with infection cocktails containing 1 mL culture medium, 3 μL Polybrene, and 0.9 μL lentiviral particles. Fifty percent medium replacement preceded addition of the infection mixture. A549-dCas9-VPR cells were seeded into T75 flasks and then infected with the library lentiviruses at a multiplicity of infection (MOI) of 0.1. Complete medium exchange occurred at 12 h post-transduction, succeeded by puromycin selection medium (2 μg/mL) application at 48 h for 4-day antibiotic resistance selection. The low-density protocol utilized an acute 8 Gy ^60^Co γ-ray irradiation delivered in a single dose to the treatment groups, after which control cells underwent subculturing on days 5 and 10 while irradiated cells were maintained without passaging through biweekly medium replacement for 14 days, culminating in day 14 cell harvest and cryopreservation at −80 °C prior to downstream genomic analyses.

### 4.3. Illumina Sequencing of sgRNAs

Genomic DNA samples from day 0 background control cells and day 14 experimental cells were submitted to Genewiz (Suzhou, China) for nucleic acid quality control. Qualified samples (2–3 μL genomic DNA) underwent sgRNA sequence amplification and subsequent library preparation, followed by high-throughput sequencing on the Illumina platform to generate raw sequencing data (FASTQ-formatted files containing gRNA core sequences). Sequencing outputs were subjected to data quantification and quality assessment prior to bioinformatic analysis, with sgRNA enrichment/depletion profiling performed to identify radiation-responsive genetic elements based on differential read abundance. Experiments were conducted with two independent biological replicates.

### 4.4. Construction of Overexpression Cell Lines

Stable overexpression cell lines for LOC401312 and CPS1 were generated using the PiggyBac transposon system through co-transfection of PiggyBac transposase plasmid PB210PA-1 and PiggyBac Dual Promoter plasmid PB514B-2 (System Biosciences, CA, USA) into A549 and H460 cell lines according to the manufacturer-recommended ratios. Following transfection, puromycin selection was implemented to establish genomically integrated cell populations, with transposase-mediated integration efficiency validated through antibiotic resistance profiling and subsequent target gene expression verification.

### 4.5. Transcriptome Sequencing

Transcriptomic sequencing with three independent biological replicates compared LOC401312-overexpressing to control cells. Total RNA was extracted using TRIzol reagent (Invitrogen, Carlsbad, CA, USA) according to the manufacturer’s procedure. Then, 10 μg of total RNA was extracted according to the RNA-Seq sample preparation kit for constructing a sequencing library. RNA libraries were prepared for sequencing using standard Illumina protocols. The cDNA libraries constructed from the pooled RNA were sequenced on the Illumina Novaseq TM 6000 sequencing platform. Using the Illumina paired-end RNA-seq approach, we sequenced the transcriptome, generating a total of million 2 × 150 bp paired-end reads. To get high quality clean reads, reads were further filtered by Cutadapt (https://cutadapt.readthedocs.io/en/stable/, version:cutadapt-1.9, accessed on 1 November 2022). The parameters were as follows: (1) removing reads containing adapters; (2) removing reads containing polyA and polyG; (3) removing reads containing more than 5% of unknown nucleotides (N); and (4) removing low quality reads containing more than 20% of low quality (Q-value ≤ 20) bases. Then, sequence quality was verified using FastQC (http://www.bioinformatics.babraham.ac.uk/projects/fastqc/, 0.11.9, accessed on 1 November 2022), including the Q20, Q30 and GC content of the clean data.

### 4.6. Colony Formation Assay

Overexpression and control cells were plated at varying densities in 6-well plates and subjected to ionizing radiation (IR) treatment at 0, 2, 4, 6, and 8 Gy doses 18 h post-seeding, followed by 10–14 days of culture with medium replacement every 48 h until macroscopic colony formation. Fixed cells were processed through sequential PBS washes, methanol fixation (20 min), and 0.2% crystal violet staining (5 min) prior to destaining and imaging. Colony quantification was performed using standardized counting criteria, with only colonies containing ≥50 cells considered valid for statistical analysis of radiation response profiles.

### 4.7. Cellular Viability Assay (CCK-8)

Following trypsinization, cells were plated at 1.5 × 10^3^ cells/well in 96-well plates and cultured for 18 h prior to exposure to ionizing radiation. Cellular proliferation was assessed at 0, 24, 48, and 72 h post-irradiation using CCK-8 assays prepared with fresh medium and CCK-8 reagent (Dojindo, Kumamoto, Kumamoto Prefecture, Japan) (9:1 *v*/*v*). Original medium was replaced with 100 µL assay solution followed by 1.5 h incubation (37 °C, 5% CO_2_), with subsequent optical density measurements at 450 nm using a spectrophotometric microplate reader. Survival curves were generated based on triplicate absorbance values normalized to non-irradiated controls.

### 4.8. RNA Extraction and RT-qPCR

Total RNA was extracted using TRIzol reagent (Sigma–Aldrich, St. Louis, MO, USA), with the RNA concentration and purity quantified via a K5600 ultra-micro spectrophotometer (Kaiao, Beijing, China), where samples demonstrating OD260/OD280 ratios between 1.8–2.0 were considered suitable for downstream applications. First-strand cDNA synthesis was performed using the HiScript III RT SuperMix reverse transcription kit (Vazyme, Nanjing, China) following the manufacturer’s protocol. Quantitative real-time PCR (qRT-PCR) analysis was conducted on the MX3000P system (Agilent, Santa Clara, CA, USA) using Applied Biosystems™ SYBR™ Green Master Mix (Applied Biosystems, Waltham, MA, USA) with thermal cycling parameters optimized per the manufacturer’s specifications. Relative expression levels of lncRNAs and mRNAs were calculated using the 2^−ΔΔCt^ method normalized to GAPDH as an endogenous control, with primer sequences for reverse transcription and amplification detailed in Appendix A.

### 4.9. Western Blot Analysis

Cellular proteins were extracted using RIPA lysis buffer (Beyotime, Shanghai, China) and quantified via a BCA assay kit (Thermo Fisher Scientific, Waltham, MA, USA), with protein concentrations normalized to 1 μg/μL using loading buffer prior to SDS-PAGE separation and subsequent electroblotting onto nitrocellulose membranes. Membranes were blocked with 8% non-fat milk for 2 h at room temperature, followed by overnight incubation at 4 °C with primary antibodies diluted in blocking solution: anti-GAPDH (1:5000, AC002, ABclonal, Wuhan, China), anti-β-tubulin (1:10,000, 10094-1-AP, Proteintech, Wuhan, China), anti-CPS1 (1:1000, ab18197, Abcam, Cambridge, UK), anti-ATM (1:1000, Y170, Abcam, Cambridge, UK), anti-phospho-ATM-Ser1981 (1:1000, ab81292, Abcam, Cambridge, UK) and anti-γH2AX(1:500, 05-636, Millipore, Billerica, MA, USA). After three TBST washes, membranes were incubated with HRP-conjugated secondary antibodies (1:5000, RGAR001/RGAM001, Proteintech, Wuhan, China) for 1 h at room temperature prior to chemiluminescent detection using ECL substrate (Millipore, Billerica, MA, USA).

### 4.10. Immunofluorescence Assay

Cells were fixed with 4% paraformaldehyde for 20 min, permeabilized with 0.5% Triton X-100 for 30 min, and blocked with 3% bovine serum albumin (BSA) for 1 h at room temperature, followed by three gentle PBS washes. Primary antibodies diluted in PBS (anti-γH2AX, 1:500, 05-636, Millipore, Billerica, MA, USA) were applied (500 μL/well) and incubated overnight at 4 °C under light-protected conditions. After three PBS washes, cells were incubated with fluorescent secondary antibodies (CoraLite^®^ Plus 488-Goat Anti-Rabbit, 1:500, RGAR002; CoraLite^®^ Plus 594-Goat Anti-Mouse, 1:500, RGAM004; Proteintech, Wuhan, China) for 1 h in darkness. Nuclei were counterstained with DAPI (1 μg/mL) for 20 min at room temperature, followed by three final PBS rinses prior to confocal microscopy imaging (Zeiss, Oberkochen, Germany) using standardized exposure settings for cross-sample comparability. Quantification of γ-H2AX foci in immunofluorescence images was conducted using ImageJ software (version 1.54p), with a positivity threshold defined as >20 foci per nucleus.

### 4.11. Comet Assay

Comet assays were performed using a commercially available kit (C2041S, Beyotime, Shanghai, China) following the manufacturer’s protocol, whereby 1 × 10^6^ cells/mL suspension was mixed with low-melting-point agarose at a 1:7.5 ratio and immediately embedded onto microscope slides pre-coated with normal-melting-point agarose, followed by 10 min solidification at 4 °C under light-protected conditions; the immobilized samples were subsequently incubated in pre-chilled lysis buffer (4 °C, dark), washed with PBS, electrophoresed in a horizontal chamber (1 V/cm for 25 min), neutralized in alkaline buffer solution, stained with 20 μL propidium iodide solution for 10–20 min in darkness, and finally visualized under a fluorescence microscope (Zeiss, Oberkochen, Germany) for quantitative analysis with CASP software (version 1.2.3b1, CaspLab), with a minimum of 20 randomly selected cells per sample being evaluated.

## 5. Conclusions

In summary, our study identified LOC401312 as a radiosensitizing lncRNA through a CRISPRa screen targeting functional lncRNAs, demonstrating that its radiation-sensitizing effect in lung cancer cells is mediated via upregulation of CPS1 and subsequent downregulation of the DNA damage repair factor ATM. These findings advance our understanding of lncRNA-mediated regulation of radiation sensitivity and provide both a mechanistic framework and actionable targets for enhancing the therapeutic efficacy of radiotherapy in lung cancer through targeted modulation of the LOC401312–CPS1 axis.

## Figures and Tables

**Figure 1 ijms-26-05865-f001:**
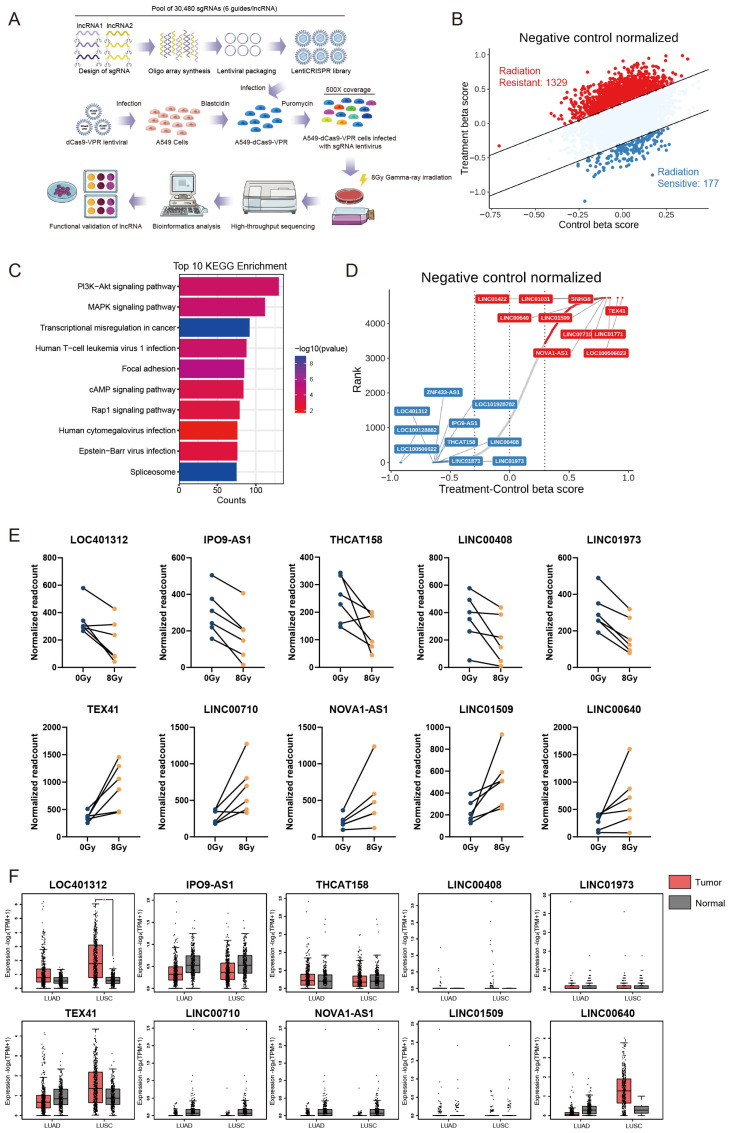
Systematic Identification of Radiation-Responsive lncRNAs via CRISPRa Screening in NSCLC. (**A**) Schematic of the CRISPRa high-throughput library screening strategy for identifying radiation-responsive lncRNAs in A549 cells, comprising 30,480 sgRNAs targeting 4766 lncRNAs (6 sgRNAs per gene) with >500× coverage per sgRNA. (**B**) Scatter plot of Delta-beta scores for screened lncRNAs. (**C**) KEGG pathway analysis and protein interaction network of candidate lncRNAs predicted using the LncSEA database (http://bio.liclab.net/LncSEA, accessed on 1 November 2021). (**D**) Delta-beta score ranking and differential gene visualization of identified lncRNAs. (**E**) Enrichment analysis of sgRNAs targeting lncRNAs, showing significant depletion of all six sgRNAs (normalized read counts shown). (**F**) Differential lncRNAs expression in normal tissues versus lung adenocarcinoma (LUAD) and squamous cell carcinoma (LUSC) clinical samples using the GEPIA2 database (http://gepia2.cancer-pku.cn, accessed on 10 June 2025). * *p* < 0.05 by unpaired two-tailed Student’s *t*-test.

**Figure 2 ijms-26-05865-f002:**
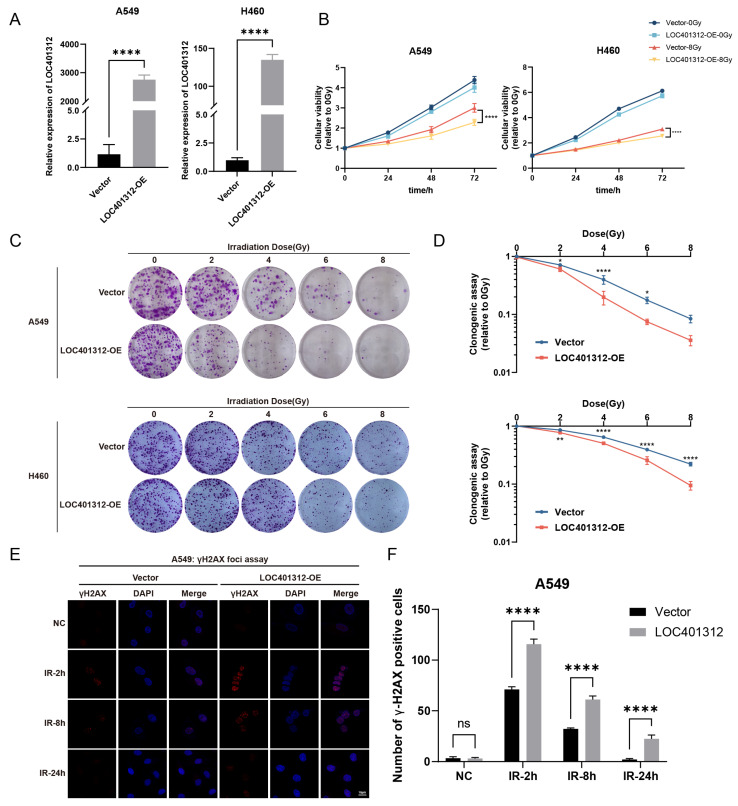
LOC401312 Validated as a Radiosensitizing lncRNA in Lung Cancer Cells. (**A**) LOC401312 overexpression validation. The RT-qPCR results were normalized against GAPDH expression. Data are presented as mean ± SD of three biologically independent replicates. **** *p* < 0.0001 by unpaired two-tailed Student’s *t*-test. (**B**) CCK-8 cellular viability curves of LOC401312-overexpressing cells post 8 Gy irradiation. Data are presented as mean ± SD of five biologically independent replicates. **** *p* < 0.0001 by two-way ANOVA. (**C**) Clonogenic survival assays at 0, 2, 4, 6, and 8 Gy (14-day post-irradiation, *n* = 3 biological replicates, the diameter of each individual image is 35 mm). (**D**) Quantification of clonogenic survival rates. Data are presented as mean ± SD * *p* < 0.05, ** *p* < 0.01, **** *p* < 0.0001 by two-way ANOVA. (**E**) Immunofluorescence imaging of γH2AX foci in LOC401312-overexpressing cells at 2 h, 8 h and 24 h post 8 Gy irradiation. Representative images of three independent experiments (*n* = 3) are shown (blue—DAPI, red—γH2AX. Scale bar: white, 10 μm). (**F**) Statistical analysis of number of γH2AX positive foci per cell. **** *p* < 0.0001 by two-way ANOVA. ns, not significant.

**Figure 3 ijms-26-05865-f003:**
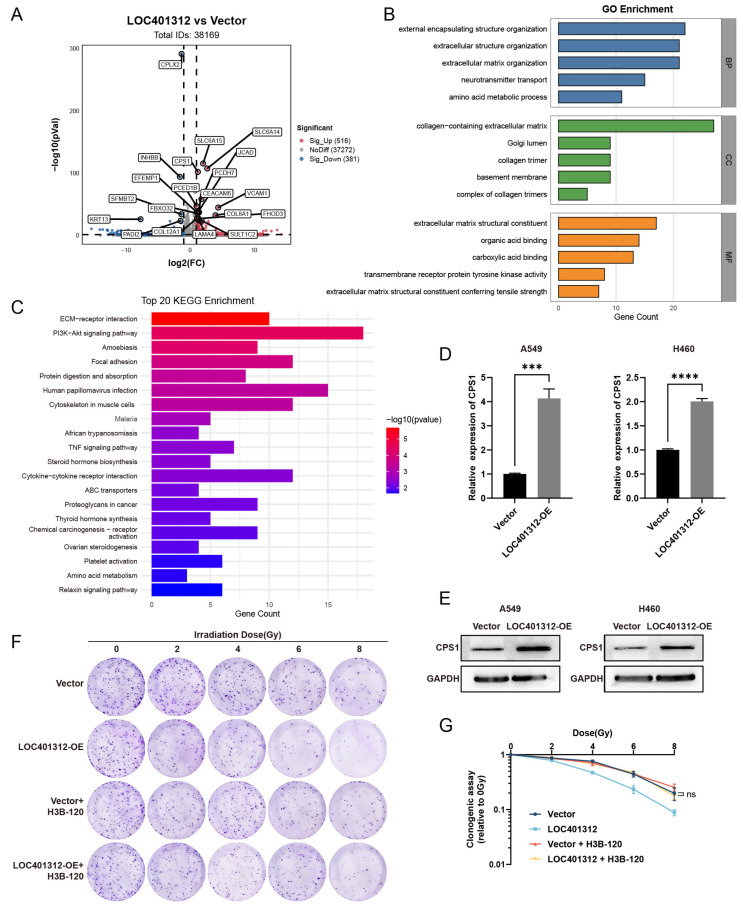
Transcriptomic Profiling of LOC401312-Overexpressing A549 Cells Identifies CPS1 as a Functional Mediator of Radiation Sensitivity. (**A**) Volcano plots show the -log normalized *p*-value and log2 fold change of transcriptomic profiling in LOC401312-overexpressing A549 cells compared to the vector control. The upregulated and downregulated DEGs are red and blue, respectively, and the undifferentiated genes expressed in both groups are denoted in gray. (**B**,**C**) GO biological process (**B**) and KEGG (**C**) analysis of DEGs in (**A**). (**D**) qRT-PCR validation of CPS1 mRNA upregulation. The RT-qPCR results were normalized against GAPDH expression. Data are presented as mean ± SD of three biologically independent replicates. *** *p* < 0.001, **** *p* < 0.0001 by unpaired two-tailed Student’s *t*-test. (**E**) Western blot confirmation of CPS1 protein induction. (**F**) Clonogenic survival assays of LOC401312-overexpressing cells irradiated with 0–8 Gy, with 10 μM H3B-120 (CPS1 inhibitor) pretreatment for 48 h (14-day post-irradiation, *n* = 3 biological replicates, the diameter of each individual image is 35 mm). (**G**) Survival fraction quantification showing abolished radiation sensitization by H3B-120; ns, not significant.

**Figure 4 ijms-26-05865-f004:**
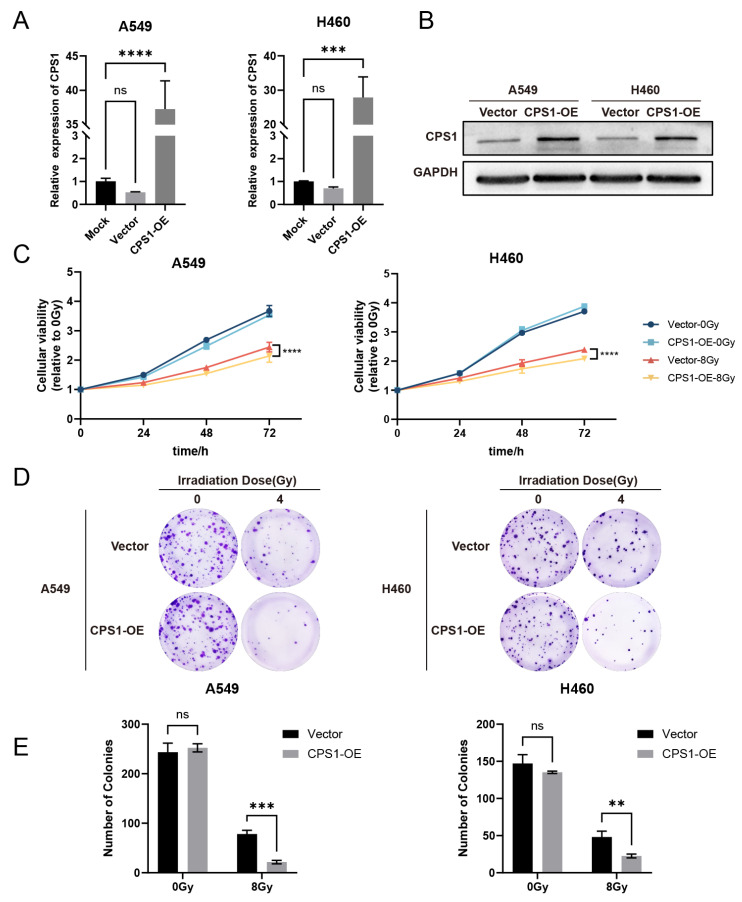
CPS1 overexpression induces radiation sensitivity in lung cancer cells. (**A**) qRT-PCR validation of CPS1 mRNA upregulation. The RT-qPCR results were normalized against GAPDH expression. Data are presented as mean ± SD of three biologically independent replicates. *** *p* < 0.001, **** *p* < 0.0001 by one-way ANOVA. (**B**) Western blot analysis of CPS1 protein levels. (**C**) CCK-8 cellular viability curves of CPS1-overexpressing cells post 8 Gy irradiation. Data are presented as mean ± SD of five biologically independent replicates. **** *p* < 0.0001 by two-way ANOVA. (**D**) Representative clonogenic survival images at 4 Gy (14-day post-irradiation, *n* = 3 biological replicates, the diameter of each individual image is 35 mm). (**E**) Quantification of number of colonies. Data are presented as mean ± SD. ** *p* < 0.01, *** *p* < 0.001 by two-way ANOVA. ns, not significant.

**Figure 5 ijms-26-05865-f005:**
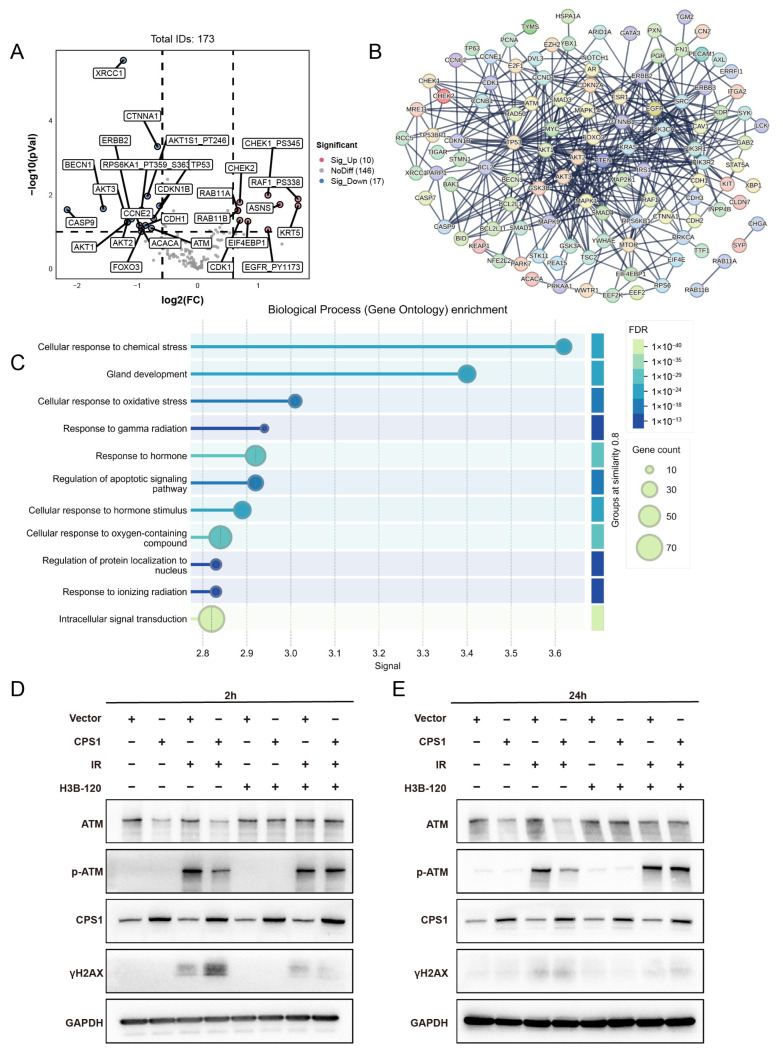
Co-expression Network Analysis of CPS1 in Lung Cancer Identifies DNA Damage Repair Modulation as a Mechanism Governing Ionizing Radiation Sensitivity. (**A**) Volcano plot of CPS1 co-expressed genes (*n* = 173) identified through cBioPortal analysis of TCGA lung cancer cohorts. (**B**) Protein–protein interaction network of CPS1-associated genes. (**C**) Gene Ontology biological process (GO-BP) enrichment analysis of CPS1-co-expressed genes. (**D**,**E**) Cells were pretreated with 10 μM CPS1 inhibitor H3B-120 for 48 h preceding 8 Gy ionizing radiation exposure, with protein lysates harvested at specified post-irradiation time points and subjected to Western blot analysis. (**F**) Immunofluorescence imaging of γH2AX foci in CPS1-overexpressing cells at 2 h, 8 h and 24 h post 8 Gy irradiation. Representative images of three independent experiments (*n* = 3) are shown (blue—DAPI, red—γH2A.X. Scale bar: white, 10 μm). (**G**) Statistical analysis of number of γH2AX positive foci per cell. ** *p* < 0.01, **** *p* < 0.0001 by two-way ANOVA. ns, not significant. (**H**) Representative comet assay images of CPS1-overexpressing versus vector-control A549 cells harvested 8 h post-8 Gy ionizing radiation, illustrating DNA damage profiles across three biologically independent experiments (scale bar: white, 100 μm). (**I**) Statistical analysis of tail DNA moment by comet assay. **** *p* < 0.0001 by two-way ANOVA. ns, not significant.

## Data Availability

The data presented in this study are available upon request.

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
