# Peer review of "Long Non-Coding RNA LOC401312 Induces Radiosensitivity Through Upregulation of CPS1 in Non-Small Cell Lung Cancer"

_ijms, 2025, doi:10.3390/ijms26125865_

Round 1
Reviewer 1 Report
Comments and Suggestions for Authors
In this study, the authors established the LOC401312-CPS1-ATM/XRCC1 axis as a previously unrecognized regulatory network governing radiation sensitivity in NSCLC. This study has a reasonable design, reliable data, which can effectively support the conclusions. However, the manuscript needs to be revised further for some questions. Some suggestions are listed as follows.
- Provide additional information on the biological replicates of the screening experiment and validate the enrichment trend of key sgRNAs through independent biological replicates
- Supplement CPS1 gene knockout (CRISPR-Cas9 or siRNA) experiments to verify whether its deletion weakens LOC401312 mediated radiosensitivity.
- The molecular details of the ATM/XRCC1 regulatory mechanism are incomplete. For example, how does CPS1 inhibit ATM phosphorylation? Does it involve the accumulation of metabolites (such as carbamoyl phosphate) or changes in ROS levels?
- Provide a zoomed-in image of the KEGG pathway in Fig 1C (such as the first 5 enriched pathways) and list the specific pathway names and p-values in the caption.
- In discussion section, combining TCGA data, analyze the correlation between LOC401312/CPS1 expression and radiotherapy prognosis in lung cancer patients. Discuss the potential synergistic effects of CPS1 inhibitors in combination with radiotherapy.
Reviewer 2 Report
Comments and Suggestions for Authors
Zhengyue Cao et al. submitted an interesting work about the effect of LOC401312 on NSCLC. The topic was of a certain significance nowadays, and might arouse some interest from the scientific community. The reviewer suggested a Minor Revision for this submission. Detailed comments:
- The logical adherence between the first two paragraphs of the Introduction was poor. Please redesign the introductive logic.
- Could Figure 2B and Figure 4C be converted into cell viability curves?
- For the GO analysis, please also provide the analysis of cellular components and molecular functions.
- A separate Conclusion Section should be added.
Reviewer 3 Report
Comments and Suggestions for Authors
This is a well-written and well-designed study that explores the role of the lncRNA LOC401312 in enhancing radiosensitivity in non-small cell lung cancer (NSCLC) via upregulation of CPS1. This work presents original findings using a comprehensive and modern set of methodologies, including CRISPRa screening, transcriptomic profiling, and multiple in vitro assays. The topic is relevant, and the results are timely and contribute meaningfully to the fields of cancer biology and radiotherapy enhancement.
The introduction provides a strong overview of the role of long non-coding RNAs (lncRNAs) in gene regulation and their relevance in the radiation response. The state of the art is well summarized, with appropriate references to key players such as GAS5, UCA1, and LINC01134. The rationale for using CRISPRa screening to identify novel radiosensitizing lncRNAs is clearly stated. However, it would be helpful to briefly introduce CPS1 earlier in the introduction since it plays a central role in the rest of the paper. Regarding the introductory section, please include 1–2 lines about CPS1's canonical role and its relevance in cancer or radiation biology earlier in the introduction to guide the reader better.
Moreover, the methods are clearly written and detailed enough to allow reproducibility. The combination of CRISPRa screening, transcriptomic profiling, functional validation (qPCR, western blotting, clonogenic assays), and bioinformatics is appropriate. However, a few points could be clarified:
- Please confirm whether the sgRNA library used was genome-wide or focused on lncRNAs only.
- It would be useful to indicate how many independent biological replicates were used for transcriptomic analysis.
- The methods section for Western blot includes both β-Tubulin and GAPDH as loading controls; please clarify whether both were used simultaneously.
Furthermore, the results are logically organized and well-supported with figures. LOC401312 is convincingly demonstrated to sensitize NSCLC cells to ionizing radiation. Additionally, the downstream identification of CPS1 as a transcriptional target, and its role in modulating the DNA damage response via ATM and XRCC1, is a strength of the study. Suggestion: please report fold-change values (and p-values) for ATM and XRCC1 downregulation explicitly in the text, not just in the figures. Also, clarify in the results whether CPS1 overexpression alone can induce radiosensitivity without LOC401312.
Also, the discussion section integrates the findings nicely and places them within the broader context of lncRNA and DDR biology. The section addressing the dual role of CPS1 in radiosensitization and DNA repair modulation is especially interesting and novel. However, there are a few overstatements that may need refinement:
- Please avoid the phrase “this study provides the first demonstration…” unless a comprehensive literature review supports it. You may rephrase as “to our knowledge” or “based on current literature.”
- The potential clinical implications of CPS1 as a therapeutic target or biomarker are compelling but could be discussed with a more cautious tone.
- Consider discussing any limitations of the study, such as lack of in vivo validation or patient-derived models.
The conclusions are supported by the data and provide a forward-looking perspective. The study does a good job of highlighting a novel regulatory axis that could be therapeutically relevant. However, it may be useful to reinforce that the mechanistic link between LOC401312 and CPS1 expression (e.g., direct transcriptional activation) remains to be fully elucidated.
Finally, a few minor edits could improve flow and clarity, such as:
Line 34: “yet only 2% encodes proteins” for “yet only 2% of it encodes proteins”
Line 259: “This section may be divided by subheadings…” seems like a leftover template text and should be deleted.
Use “phosphorylation of ATM at Ser1981” instead of “ATM phosphorylation at Ser1981” for clarity in some parts of the text.
Reviewer 4 Report
Comments and Suggestions for Authors
The authors identified long non-coding RNA LOC401312, which may be a radiosensitivity regulator in non-small cell lung cancer cell lines A549 and H460 using CRISPR technology. Furthermore, LOC401312 regulates CPS1 expression and overexpression of CPS1 suppresses the phosphorylation of ATM kinase and XRCC1 protein levels. The assessment methods of radiosensitivity were growth assay, colony formation assay, and gamma-H2AX (i.e., DNA double strand break). I think this manuscript makes occasional leaps of logic and the experiments require further expansion and clarification.
Major comments
- In introduction, the second paragraph is unclear. Although specific molecules are mentioned in the latter half, what do the authors ultimately want to say in the introduction of this study? If you are going to write about lncRNA in radiosensitivity, it would be easier to understand if you wrote whether it is involved in DNA repair or antioxidant activity. Third paragraph of introduction is lengthy, making it unclear what the authors are trying to say. It would be better to explain CRISPRa at the beginning and then add other supplementary elements. In addition, the fourth paragraph, if you write the sentence “Despite growing recognition…radiation response pathways,” I recommend switching the second and third paragraphs and providing examples of research on lncRNA.
- Basically, ATM is responsible for DNA double-strand break repair, while XRCC1 is responsible for DNA single-strand break repair, and they are thought to be unrelated. I don’t think there is any need to link these two molecules, so why don’t you change your consideration? If you want to link them, further experiments are required.
- I think the reason why residual DNA damage (i.e., 8 h after irradiation) is significantly high compared with vector (Fig. 2E and 2F) is because there is a lot of initial DNA damage, and the repair kinetics itself does not seem to change. How many hours does it take to return to a steady state? For example, it would be better to measure it later, after 24–48 h. The author should consider that molecules involved in DNA damage repair change over time.
- Although the overexpression of CPS1 affected the radiosensitivity, the expression of CPS1 itself was not changed after 8 Gy irradiation (Fig. 5D; the authors did not quantify). This suggests that CPS1 does not and is not regulated in response to irradiation. It is necessary to clarify the quantitative expression of CPS1 under irradiation and dose (time) dependence. Besides, the authors should clarify the time of harvest after irradiation.
- Is CPS1 related to the prognosis of lung cancer patients in TCGA dataset analysis? The authors suggested CPS1 emerges as a dual-functional candidate in lung cancer patients, and one side is a diagnostic biomarker. The authors can easily analyze it using a tool like UCSC Xena.
Specific comments
- Introduction L:48, what does “cellular constituents” mean? This sentence is not specified. In radiation biology, it should be DNA.
- Introduction L:63, “DNA damage checkpoints”: This should be cell cycle checkpoints.
- Results L: 118, “radiation-associated”: This should be radiosensitivity-associated.
- Results L:123-124, “significant downregulation…lung adenocarcinoma”: In lung adenocarcinoma, not significant downregulation of LOC401312.
- Results L:148-149, “which exceeds the threshold of cellular repair mechanisms”: Please explain how you defined it.
- Figure 2D and 3G, surviving fraction should be on a logarithmic scale.
- Results L:253-255, “This work collectively…distinct biological pathways”: This sentence is unclear. Is this limited to A549?
- Results L:267-269, draft comments? Remove them.
- Discussion L:273-275, this is an exaggeration because this study only briefly investigated the DDR pathway.
- Discussion L:283-284, the results of this paper indicate an increase in the number of early DNA damage events.
- Discussion L:313, “metabolic-epigenetic regulator of DNA damage response”: What is the basis for this? I think it cannot be stated from the experimental results of this study.
- Discussion L;318-319, Please rephrase the metabolic-epigenetic crosstalk as it is not the focus of this study.
- Materials and Methods, “4.9 Immunofluorescence Assay”: This section incomplete.
Round 2
Reviewer 4 Report
Comments and Suggestions for Authors
I would like to thank the authors for addressing all my comments. The manuscript has been improved and is more impactful. However, microscopic images of γH2AX should be brightened for easy visual observation.
Author Response
Comment : The manuscript has been improved and is more impactful. However, microscopic images of γH2AX should be brightened for easy visual observation.
Response : We sincerely appreciate the reviewer's insightful suggestions. We have optimized the brightness of all γH2AX immunofluorescence micrographs (updated Figure 2E & 5F) to ensure enhanced visualization of radiation-induced foci, and these revised images have been incorporated into the manuscript.